# Concentration of risk measures:
# A Wasserstein distance approach

**Sanjay P. Bhat**
Tata Consultancy Services Limited
Hyderabad, Telangana, India
sanjay.bhat@tcs.com

**Prashanth L.A.**
Department of Computer Science and Engineering
Indian Institute of Technology Madras, India
prashla@cse.iitm.ac.in
[*]

## Abstract

Known finite-sample concentration bounds for the Wasserstein distance between the empirical and true distribution of a random variable are used to derive a two-sided concentration bound for the error between the true conditional value-at-risk (CVaR) of a (possibly unbounded) random variable and a standard estimate of its CVaR computed from an i.i.d. sample. The bound applies under fairly general assumptions on the random variable, and improves upon previous bounds which were either one sided, or applied only to bounded random variables. Specializations of the bound to sub-Gaussian and sub-exponential random variables are also derived. Using a different proof technique, the results are extended to the class of spectral risk measures having a bounded risk spectrum. A similar procedure is followed to derive concentration bounds for the error between the true and estimated Cumulative Prospect Theory (CPT) value of a random variable, in cases where the random variable is bounded or sub-Gaussian. These bounds are shown to match a known bound in the bounded case, and improve upon the known bound in the sub-Gaussian case. The usefulness of the bounds is illustrated through an algorithm, and corresponding regret bound for a stochastic bandit problem, where the underlying risk measure to be optimized is CVaR.

## 1 Introduction

Conditional Value-at-Risk (CVaR) and cumulative prospect theory (CPT) value are two popular risk measures. CVaR is popular in financial applications, where it is necessary to minimize the worst-case losses, say in a portfolio optimization context. CVaR is a special instance of the class of spectral risk measures [Acerbi, 2002]. CVaR is an appealing risk measure because it is coherent [Artzner et al., 1999], and spectral risk measures retain this property. CPT value is a risk measure, proposed by Tversky and Kahnemann, that is useful for modeling human preferences. The central premise in risk-sensitive optimization is that the expected value is not an appealing objective in several practical applications, and it is necessary to incorporate some notion of risk in the optimization process. The reader is referred to extensive literature on risk-sensitive optimization, in particular, the shortcomings of the expected value - cf. [Allais, 1953, Ellsberg, 1961, Kahneman and Tversky, 1979, Rockafellar and Uryasev, 2000].

In practical applications, the information about the underlying distribution is typically unavailable. However, one can often obtain samples from the distribution, and the aim is to estimate the chosen risk measure using these samples. We consider this problem of estimation in the context of three risk measures: CVaR, a general spectral risk measure, and CPT-value. For each of the three risk measures, we examine the estimator obtained by applying the risk measure to the empirical

---

[*]Supported in part by a DST grant under the ECRA program.

distribution constructed from an i.i.d. sample. In the case of CVaR and CPT value, the estimators obtained in this way are already available in the literature. Our goal is to derive concentration bounds for estimators of all three risk measures, and we achieve this in a novel manner by relating the estimation error to the Wasserstein distance between the empirical and true distributions, and then using known concentration bounds for the latter. We summarize our contributions below, which apply when the underlying distribution has a bounded exponential moment, or a higher-order moment. Sub-Gaussian distributions are a popular class that satisfy the former condition, while the latter includes sub-exponential distributions.

**(1)** For the case of CVaR, we provide a two-sided concentration bound for both classes of distributions mentioned above. In particular, for the special case of sub-Gaussian distributions, our tail bound is of the order $O\left(\exp\left(-cn\epsilon^2\right)\right)$, where $n$ is the number of samples, $\epsilon$ is the accuracy parameter, and $c$ is a universal constant. Our bound matches the rate obtained for distributions with bounded support in [Brown, 2007], and features improved dependence on $\epsilon$ as compared to the one derived for sub-Gaussian distributions in [Kolla et al., 2019b]. Further, unlike the latter work, we provide two-sided concentration bounds for CVaR estimation. Similar bounds are shown to hold for any spectral risk measure having a bounded risk spectrum.

**(2)** For the case of CPT-value, we obtain an order $O\left(\exp\left(-cn\epsilon^2\right)\right)$ for the case of distributions with bounded support, matching the rate in [Cheng et al., 2018]. For the case of sub-Gaussian distributions, we provide a bound that has an improved dependence on the number of samples $n$, as compared to the corresponding bound derived by [Cheng et al., 2018].

**(3)** As a minor contribution, our concentration bounds open avenues for bandit applications, and we illustrate this claim by considering a risk-sensitive bandit setting, with CVaR as the underlying risk measure. For this bandit problem with underlying arms' distribution assumed to be sub-Gaussian, we derive a regret bound using the CVaR concentration bound mentioned above. Previous works (cf. [Galichet et al., 2013]) consider CVaR optimization in a bandit context, with arms' distributions having bounded support.

Since CVaR and spectral risk measures are weighted averages of the underlying distribution quantiles, a natural alternative to a Wasserstein-distance-based approach is to employ concentration results for quantiles such as in Kolla et al. [2019b]. While such an approach can provide bounds with better constants, the resulting bounds also involve distribution-dependent quantities (see Kolla et al. [2019b], for instance), and require different proofs for sub-Gaussian and sub-exponential random variables. In contrast, our approach provides a unified method of proof.

The rest of the paper is organized as follows: In Section 2, we cover background material that includes Wasserstein distance, and sub-Gaussian and sub-exponential distributions. In Section 3–5, we present concentration bounds for CVaR, spectral risk measures and CPT-value estimation, respectively. In Section 6, we discuss a bandit application, and finally, in Section 7, we provide the concluding remarks. The proofs of all the claims in Sections 3–5 are given in the supplementary material.

## 2   Wasserstein Distance

In this section, we introduce the notion of Wasserstein distance, a popular metric for measuring the proximity between two distributions. The reader is referred to Chapter 6 of [Villani, 2008] for a detailed introduction.

Given two cumulative distribution functions (CDFs) $F_1$ and $F_2$ on $\mathbb{R}$, let $\Gamma(F_1, F_2)$ denote the set of all joint distributions on $\mathbb{R}^2$ having $F_1$ and $F_2$ as marginals.

**Definition 1.** *Given two CDFs $F_1$ and $F_2$ on $\mathbb{R}$, the Wasserstein distance between them is defined by*

$$W_1(F_1, F_2) = \left[\inf_{F \in \Gamma(F_1, F_2)} \int_{\mathbb{R}^2} |x - y| dF(x, y)\right]. \tag{1}$$

Given $L > 0$ and $p > 0$, a function $f : \mathbb{R} \to \mathbb{R}$ is $L$-Hölder of order $p$ if $|f(x) - f(y)| \leq L|x - y|^p$ for all $x, y \in \mathbb{R}$. The function $f : \mathbb{R} \to \mathbb{R}$ is $L$-Lipschitz if it is $L$-Hölder of order 1. Finally, if $F$ is a CDF on $\mathbb{R}$, we define the generalized inverse $F^{-1} : [0, 1] \to \mathbb{R}$ of $F$ by $F^{-1}(\beta) = \inf\{x \in \mathbb{R} : F(x) \geq \beta\}$. In the case where $F$ is strictly increasing and continuous, $F^{-1}$ equals the usual inverse of a bijective function.

The Wasserstein distance between the CDFs $F_1$ and $F_2$ of two random variables $X$ and $Y$, respectively, may be alternatively written as follows:

$$\sup |\mathbb{E}\left(f(X)\right) - \mathbb{E}(f(Y))| = W_1(F_1, F_2) = \int_{-\infty}^{\infty} |F_1(s) - F_2(s)| \mathrm{d}s = \int_0^1 |F_1^{-1}(\beta) - F_2^{-1}(\beta)| \mathrm{d}\beta, \tag{2}$$

where the supremum in (2) is over all functions $f : \mathbb{R} \to \mathbb{R}$ that are 1-Lipschitz. Equation (2) is stated and proved as a lemma in Bhat and Prashanth [2019].

The results that we provide in this paper pertain to the case where a r.v. $X$ satisfies either an exponential moment bound or a higher-order moment bound. We make these conditions precise below.

**(C1)** There exist $\beta > 0$ and $\gamma > 0$ such that $\mathbb{E}\left(\exp\left(\gamma |X - \mu|^\beta\right)\right) < \top < \infty$, where $\mu = \mathbb{E}(X)$.

**(C2)** There exists $\beta > 0$ such that $\mathbb{E}\left(|X - \mu|^\beta\right) < \top < \infty$, where $\mu = \mathbb{E}(X)$.

We next define sub-Gaussian and sub-exponential r.v.s., which are two popular classes of unbounded r.v.s, that satisfy assumptions (C1) and (C2), respectively.

**Definition 2.** *A r.v. $X$ with mean $\mu$ is sub-Gaussian if there exists a $\sigma > 0$ such that*

$$\mathbb{E}(\exp\left(\lambda(X - \mu)\right)) \le \exp\left(\frac{\lambda^2 \sigma^2}{2}\right) \text{ for any } \lambda \in \mathbb{R}.$$

A sub-Gaussian r.v. $X$ with mean $\mu$ satisfies (see items (II) and (IV) in Theorem 2.1 of [Wainwright, 2019] for a proof)

$$\mathbb{E}\left(\exp\left(\frac{(X - \mu)^2}{4\sigma^2}\right)\right) \le \sqrt{2}, \text{ and } P(X - \mu > \eta) \le 8\exp(-\frac{\eta^2}{2\sigma^2}), \text{ for } \eta \ge 0. \tag{3}$$

The first bound above implies that sub-Gaussian r.v.s satisfy (C1) with $\beta = 2$, $\gamma = \frac{1}{4\sigma^2}$ and $\top = \sqrt{2}$. In particular, bounded r.v.s are sub-Gaussian, and satisfy (C1) with $\beta = 2$.

**Definition 3.** *Given $\sigma > 0$, a r.v. $X$ with mean $\mu$ is $\sigma$ sub-exponential if there exist non-negative parameters $\sigma$ and $b$ such that*

$$\mathbb{E}(\exp\left(\lambda(X - \mu)\right)) \le \exp\left(\frac{\lambda^2 \sigma^2}{2}\right) \text{ for any } |\lambda| < \frac{1}{b}.$$

A sub-exponential r.v. $X$ with mean $\mu$ satisfies (see items (III) and (IV) in Theorem 2.2 of [Wainwright, 2019] for a proof)

$$\sup_{k \ge 2} \left[\frac{\mathbb{E}\left[(X - \mu)^k\right]}{k!}\right]^{\frac{1}{k}} < \infty, \text{ and } \exists k_1, k_2 > 0 \text{ such that } \mathbb{P}\left(X - \mu > \eta\right) \le k_1 \exp(-k_2 \eta), \forall \eta \ge 0. \tag{4}$$

The bound (4) implies that sub-exponential r.v.s satisfy (C2) for integer values of $\beta \ge 2$.

The following result from Fournier and Guillin [2015] bounds the Wasserstein distance between the empirical distribution function (EDF) of an i.i.d. sample and the underlying CDF from which the sample is drawn. Recall that, given $X_1, \ldots, X_n$ i.i.d. samples from the distribution $F$ of a r.v. $X$, the EDF $F_n$ is defined by

$$F_n(x) = \frac{1}{n} \sum_{i=1}^n \mathbb{I}\left\{X_i \le x\right\}, \text{ for any } x \in \mathbb{R}. \tag{5}$$

**Lemma 1.** *(Wasserstein distance bound) Let $X$ be a r.v. with CDF $F$ and mean $\mu$. Suppose that either (i) $X$ satisfies (C1) with $\beta > 1$, or (ii) $X$ satisfies (C2) with $\beta > 2$. Then, for any $\epsilon \ge 0$, we have*

$$\mathbb{P}\left(W_1(F_n, F) > \epsilon\right) \le B(n, \epsilon),$$

*where, under (i),*

$$B(n, \epsilon) = C\left(\exp\left(-cn\epsilon^2\right) \mathbb{I}\left\{\epsilon \le 1\right\} + \exp\left(-cn\epsilon^\beta\right) \mathbb{I}\left\{\epsilon > 1\right\}\right),$$

*for some $C, c$ that depend on the parameters $\beta, \gamma$ and $\top$ specified in (C1); and under (ii),*

$$B(n, \epsilon) = C \left( \exp\left(-cn\epsilon^2\right) \mathbb{I}\{\epsilon \le 1\} + n \left(n\epsilon\right)^{-(\beta-\eta)/p} \mathbb{I}\{\epsilon > 1\} \right).$$

*where $\eta$ could be chosen arbitrarily from $(0, \beta)$, while $C, c$ depend on the parameters $\beta, \eta$ and $\top$ specified in (C2).*

*Proof.* The lemma follows directly by applying Theorem 2 in [Fournier and Guillin, 2015] to the random variable $X - \mu$, and noting from (2) that the Wasserstein distance remains invariant if the same constant is added to both random variables. □

## 3   Conditional Value-at-Risk

We now introduce the notion of CVaR, a risk measure that is popular in financial applications.

**Definition 4.** *The CVaR at level $\alpha \in (0, 1)$ for a r.v $X$ is defined by*

$$C_\alpha(X) = \inf_\xi \left\{ \xi + \frac{1}{(1-\alpha)} \mathbb{E}\left(X - \xi\right)^+ \right\}, \text{ where } (y)^+ = \max(y, 0).$$

It is well known (see [Rockafellar and Uryasev, 2000]) that the infimum in the definition of CVaR above is achieved for $\xi = \text{VaR}_\alpha(X)$, where $\text{VaR}_\alpha(X) = F^{-1}(\alpha)$ is the value-at-risk of the random variable $X$ at confidence level $\alpha$. Thus CVaR may also be written alternatively as given, for instance, in [Kolla et al., 2019b]. In the special case where $X$ has a continuous distribution, $C_\alpha(X)$ equals the expectation of $X$ conditioned on the event that $X$ exceeds $\text{VaR}_\alpha(X)$.

All our results below pertain to i.i.d. samples $X_1, \ldots, X_n$ drawn from the distribution of $X$. Following Brown [2007], we estimate $C_\alpha(X)$ from such a sample by

$$c_{n,\alpha} = \inf_\xi \left\{ \xi + \frac{1}{n(1-\alpha)} \sum_{i=1}^n \left(X_i - \xi\right)^+ \right\}. \tag{6}$$

We now provide a concentration bound for the empirical CVaR estimate (6), by relating the estimation error $|c_{n,\alpha} - C_\alpha(X)|$ to the Wasserstein distance between the true and empirical distribution functions, and subsequently invoking Lemma 1 that bounds the Wasserstein distance between these two distributions. The proof is given in section 5 of Bhat and Prashanth [2019].

**Proposition 1.** *Suppose $X$ either satisfies (C1) for some $\beta > 1$ or satisfies (C2) for some $\beta > 2$. Under (C1), for any $\epsilon > 0$, we have*

$$\mathbb{P}\left(|c_{n,\alpha} - C_\alpha(X)| > \epsilon\right) \le C\left[\exp\left[-cn(1-\alpha)^2\epsilon^2\right] \mathbb{I}\{\epsilon \le 1\} + \exp\left[-cn(1-\alpha)^\beta \epsilon^\beta\right] \mathbb{I}\{\epsilon > 1\}\right].$$

*Under (C2), for any $\epsilon > 0$, we have*

$$\mathbb{P}\left(|c_{n,\alpha} - C_\alpha(X)| > \epsilon\right) \le C\left[\exp\left[-cn(1-\alpha)^2\epsilon^2\right] \mathbb{I}\{\epsilon \le 1\} + n \left(n(1-\alpha)\epsilon\right)^{-(\beta-\eta)} \mathbb{I}\{\epsilon > 1\}\right].$$

*In the above, the constants $C, c$ and $\eta$ are as in Lemma 1.*

The following corollary, which specializes Proposition 1 to sub-Gaussian random r.v.s., is immediate, as sub-Gaussian random variables satisfy (C1) with $\beta = 2$.

**Corollary 1.** *For a sub-Gaussian r.v. $X$, we have that*

$$\mathbb{P}\left(|c_{n,\alpha} - C_\alpha(X)| > \epsilon\right) \le 2C \exp\left(-cn(1-\alpha)^2\epsilon^2\right), \text{ for any } \epsilon \ge 0,$$

*where $C, c$ are constants that depend on the sub-Gaussianity parameter $\sigma$.*

In terms of dependence on $n$ and $\epsilon$, the tail bound above is better than the one-sided concentration bound in [Kolla et al., 2019b]. In fact, the dependence on $n$ and $\epsilon$ matches that in the case of bounded distributions (cf. [Brown, 2007, Wang and Gao, 2010]).

The case of sub-exponential distributions can be handled by specializing the second result in Proposition 1. In particular, observing that sub-exponential distributions satisfy (C2) for any $\beta \ge 2$, and Proposition 1 requires $\beta > 2$ in case (ii), we obtain the following bound:

**Corollary 2.** *For a sub-exponential r.v. $X$, for any $\epsilon \geq 0$, we have*

$$\mathbb{P}\left(|c_{n,\alpha} - C_\alpha(X)| > \epsilon\right) \leq C\left[\exp\left[-cn(1-\alpha)^2\epsilon^2\right]\mathbb{I}\{\epsilon \leq 1\} + n\left[n(1-\alpha)\epsilon\right]^{\eta-3}\mathbb{I}\{\epsilon > 1\}\right],$$

*where $C, c$ and $\eta$ are as in Lemma 1.*

For small deviations, i.e., $\epsilon \leq 1$, the bound above is satisfactory, as the tail decay matches that of a Gaussian r.v. with constant variance. On the other hand, for large $\epsilon$, the second term exhibits polynomial decay. The latter polynomial term is not an artifact of our analysis, and instead, it relates to the rate obtained in case (ii) of Lemma 1. Sub-exponential distributions satisfy an exponential moment bound with $\beta = 1$, and for this case, the authors in [Fournier and Guillin, 2015] remark that they were not able to obtain a satisfactory concentration result. Recently, Prashanth et al. [2019] have derived an improved bound for the sub-exponential case using a technique not based on the Wasserstein distance.

## 4 Spectral risk measures

Spectral risk measures are a generalization of CVaR. Given a weighting function $\phi : [0,1] \rightarrow [0,\infty)$, the spectral risk measure $M_\phi$ associated with $\phi$ is defined by

$$M_\phi(X) = \int_0^1 \phi(\beta)F^{-1}(\beta)\mathrm{d}\beta, \tag{7}$$

where $X$ is a random variable with CDF $F$. If the weighting function, also known as the *risk spectrum*, is increasing and integrates to 1, then $M_\phi$ is a coherent risk measure like CVaR. In fact, CVaR is itself a special case of (7), with $C_\alpha(X) = M_\phi$ for the risk spectrum $\phi = (1-\alpha)^{-1}\mathbb{I}\{\beta \geq \alpha\}$ (see Acerbi [2002] and Dowd and Blake [2006] for details).

Given an i.i.d. sample $X_1, \ldots, X_n$ drawn from the CDF $F$ of a random variable $X$, a natural empirical estimate of the spectral risk measure $M_\phi(X)$ of $X$ is

$$m_{n,\phi} = \int_0^1 \phi(\beta)F_n^{-1}(\beta)\mathrm{d}\beta. \tag{8}$$

In this section, we restrict ourselves to a spectral risk measure $M_\phi$ whose associated risk spectrum $\phi$ is bounded. Specifically, we assume that $|\phi(\beta)| \leq K$ for all $\beta \in [0,1]$ for some $K > 0$. It immediately follows from (7) and (2) that, if $X$ and $Y$ are random variables with CDFs $F_1$ and $F_2$, then

$$|M_\phi(X) - M_\phi(Y)| \leq KW_1(F_1, F_2). \tag{9}$$

On noting from (8) that the empirical estimate $m_{n,\phi}$ of $M_\phi(X)$ is simply the spectral risk measure $M_\phi$ applied to a random variable whose CDF is $F_n$, we conclude from (9) that

$$|M_\phi(X) - m_{n,\phi}| \leq KW_1(F, F_n). \tag{10}$$

Equation (10) relates the estimation error $|M_\phi(X) - m_{n,\phi}|$ to the Wasserstein distance between the true and empirical CDFs of $X$. As in the case of CVaR, invoking Lemma 1 provides concentration bounds for the empirical spectral risk measure estimate (8). The detailed proof is available in Section 5 of Bhat and Prashanth [2019].

**Proposition 2.** *Suppose $X$ either satisfies (C1) for some $\beta > 1$ or satisfies (C2) for some $\beta > 2$. Let $K > 0$ and let $\phi : [0,1] \rightarrow [0,K]$ be a risk spectrum for some $K > 0$. Under (C1), for any $\epsilon > 0$, we have*

$$\mathbb{P}\left(|m_{n,\phi} - M_\phi(X)| > \epsilon\right) \leq C\left[\exp\left[-cn\left\{\frac{\epsilon}{K}\right\}^2\right]\mathbb{I}\{\epsilon \leq 1\} + \exp\left[-cn\left\{\frac{\epsilon}{K}\right\}^\beta\right]\mathbb{I}\{\epsilon > 1\}\right].$$

*Under (C2), for any $\epsilon > 0$, we have*

$$\mathbb{P}\left(|m_{n,\phi} - M_\phi(X)| > \epsilon\right) \leq C\left[\exp\left[-cn\left\{\frac{\epsilon}{K}\right\}^2\right]\mathbb{I}\{\epsilon \leq 1\} + n\left(n\left\{\frac{\epsilon}{K}\right\}\right)^{-(\beta-\eta)/p}\mathbb{I}\{\epsilon > 1\}\right].$$

*In the above, the constants $C, c$ and $\eta$ are as in Lemma 1.*

The following corollary specializing Proposition 2 to sub-Gaussian random r.v.s. is immediate, as sub-Gaussian random variables satisfy (C1) with $\beta = 2$.

**Corollary 3.** *For a sub-Gaussian r.v. $X$ and a risk spectrum as in Proposition 2, we have*

$$\mathbb{P}\left(|m_{n,\phi} - M_\phi(X)| > \epsilon\right) \leq 2C \exp\left(-cn\epsilon^2/K^2\right), \text{ for any } \epsilon \geq 0,$$

*where $C, c$ are constants that depend on $\sigma$.*

It is possible to specialize Proposition 2 to the case of sub-exponential random variables to obtain a corollary similar to Corollary 2. However, in the interests of space, we do not present it here.

Technically speaking, the concentration bounds for CVaR from Section 3 follow from the results of this section, since CVaR is a special case of a spectral risk measure. However, the proof technique of Section 3 uses a different characterization of the Wassserstein distance, and is based on a different formula for CVaR. We therefore believe that the independent proofs given for the results of Section 3 are interesting in their own right.

## 5 CPT-value estimation

For any r.v. $X$, the CPT-value is defined as

$$C(X) = \int_0^\infty w^+ \left(\mathbb{P}\left(u^+(X) > z\right)\right) dz - \int_0^\infty w^- \left(\mathbb{P}\left(u^-(X) > z\right)\right) dz, \tag{11}$$

Let us deconstruct the above definition. The functions $u^+, u^- : \mathbb{R} \to \mathbb{R}_+$ are utility functions that are assumed to be continuous, with $u^+(x) = 0$ when $x \leq 0$ and increasing otherwise, and with $u^-(x) = 0$ when $x \geq 0$ and decreasing otherwise. The utility functions capture the human inclination to play safe with gains and take risks with losses – see Fig 1. Second, $w^+, w^- : [0, 1] \to [0, 1]$ are weight functions, which are assumed to be continuous, non-decreasing and satisfy $w^+(0) = w^-(0) = 0$ and $w^+(1) = w^-(1) = 1$. The weight functions $w^+, w^-$ capture the human inclination to view probabilities in a non-linear fashion. Tversky and Kahneman [1992], Barberis [2013] (see Fig 2 from Tversky and Kahneman [1992]) recommend the following choices for $w^+$ and $w^-$, based on inference from experiments involving human subjects:

$$w^+(p) = \frac{p^{0.61}}{(p^{0.61} + (1-p)^{0.61})^{\frac{1}{0.61}}}, \text{ and } w^-(p) = \frac{p^{0.69}}{(p^{0.69} + (1-p)^{0.69})^{\frac{1}{0.69}}}. \tag{12}$$

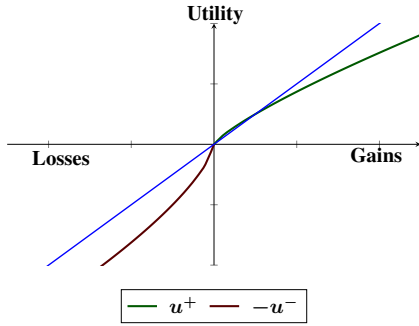

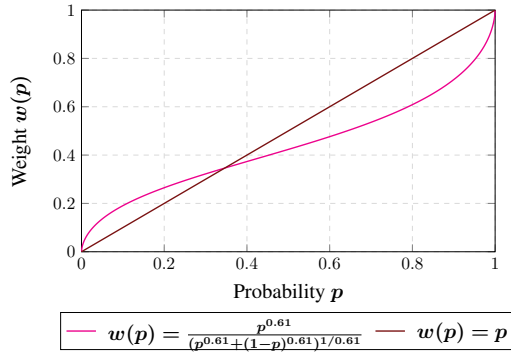

Figure 1: Utility function

Figure 2: Weight function

We now recall CPT-value estimation proposed in [Prashanth et al., 2016]. Let $X_i$, $i = 1, \ldots, n$ denote $n$ samples from the distribution of $X$. The EDF for $u^+(X)$ and $u^-(X)$, for any given real-valued functions $u^+$ and $u^-$, is defined as follows: $\hat{F}_n^+(x) = \frac{1}{n}\sum_{i=1}^n \mathbb{I}\left\{(u^+(X_i) \leq x)\right\}, \hat{F}_n^-(x) = \frac{1}{n}\sum_{i=1}^n \mathbb{I}\left\{(u^-(X_i) \leq x)\right\}.$ Using EDFs, the CPT-value is estimated as follows:

$$C_n = \int_0^\infty w^+(1 - \hat{F}_n^+(x))dx - \int_0^\infty w^-(1 - \hat{F}_n^-(x))dx. \tag{13}$$

Notice that we have substituted the complementary EDFs $\left(1 - \hat{F}_n^+(x)\right)$ and $\left(1 - \hat{F}_n^-(x)\right)$ for $\mathbb{P}\left(u^+(X) > x\right)$ and $\mathbb{P}\left(u^-(X) > x\right)$, respectively, in (11), and then performed an integration of the weight function composed with the complementary EDF. As shown in Section III of [Prashanth et al., 2016], the first and second integral in (13) can be easily computed using the order statistics $\{X_{(1)}, \ldots, X_{(n)}\}$.

For the purpose of analysis, as in [Cheng et al., 2018], we make the following assumption:

**(C3)** The weight functions $w^+, w^-$ are $L$-Hölder continuous of order $\alpha \in (0, 1)$ for some constant $L > 0$.

In this paper, we are interested in deriving a concentration bound for the estimator in (13). To put things in context, in [Cheng et al., 2018], the authors derive a concentration bound assuming that the underlying distribution has bounded support, and for this purpose, they employ the Dvoretzky-Kiefer-Wolfowitz (DKW) theorem (cf. Chapter 2 of [Wasserman, 2015]). Interestingly, we are able to provide a matching bound for the case of distributions with bounded support, using a proof technique that relates the the estimation error $|C_n - C(X)|$ to the Wasserstein distance between the empirical and true CDF, and this is the content of the proposition below (see Section 5 of Bhat and Prashanth [2019] for the proof).

**Proposition 3.** *(CPT concentration for bounded r.v.s) Let $X_1, \ldots, X_n$ be i.i.d. samples of a r.v. $X$ that is bounded a.s. in $[-T_1, T_2]$, where $T_1, T_2 \geq 0$, and at least one of $T_1, T_2$ is positive. Let $T \triangleq \max\{u^+(T_2), u^-(-T_1)\}$. Then, under (C3), we have*

$$\mathbb{P}\left(|C_n - C(X)| > \epsilon\right) \leq 2B\left(n, \left[\frac{\epsilon}{2LT^{1-\alpha}}\right]^{1/\alpha}\right), \text{ for any } \epsilon \geq 0,$$

*where $B(\cdot, \cdot)$ is as given in i) of Lemma 1 with $\beta = 2$.*

From the form for $B(\cdot, \cdot)$ in Lemma 1, it is apparent that $|C_n - C(X)| < \epsilon$ with probability $1 - \delta$, if the number of samples $n$ is of the order $O\left(1/\epsilon^{2/\alpha} \log\left(\frac{1}{\delta}\right)\right)$, for any $\delta \in (0, 1)$.

Next, we provide a CPT concentration result for the case when the underlying r.v. is unbounded, but sub-Gaussian. For this case, we consider a modified CPT value estimator based on truncation, namely,

$$\tilde{C}_n = \int_0^{\tau_n} w^+(1 - \hat{F}_n^+(z))\mathrm{d}z - \int_0^{\tau_n} w^-(1 - \hat{F}_n^-(z))\mathrm{d}z,$$

where the sample-size-dependent truncation threshold $\tau_n$ is specified in the result below. The proof is available in Bhat and Prashanth [2019].

**Proposition 4.** *(CPT concentration for sub-Gaussian r.v.s) Let $X_1, \ldots, X_n$ be i.i.d. samples from the distribution of $X$. Suppose that $u^+(X)$ and $u^-(X)$ are sub-Gaussian r.v.s with parameter $\sigma$. Set $\tau_n = \sigma\left(\sqrt{\log n} + \sqrt{\log \log n}\right)$ for all $n \geq 1$. Then, for all $n$ satisfying $\sigma\sqrt{\log \log n} > \max\left(\mathbb{E}(u^+(X)), \mathbb{E}(u^-(X))\right) + 1$, we have*

$$\mathbb{P}\left(\left|\tilde{C}_n - C(X)\right| > \epsilon\right) \leq 2C \exp\left(-cn\left(\frac{\epsilon - \frac{8L\sigma^2}{\alpha n^{\alpha/2}}}{L\sqrt{\log n}}\right)^{\frac{2}{\alpha}}\right) \text{ for every } \epsilon > \frac{8L\sigma^2}{\alpha n^{\alpha/2}},$$

*where $C, c$ are constants that depend on the sub-Gaussianity parameter $\sigma$.*

The corresponding bound provided in Proposition 3 of Cheng et al. [2018] is $\left(2ne^{-n^{\frac{\alpha}{2+\alpha}}} + 2e^{-n^{\frac{\alpha}{2+\alpha}}\left(\frac{\epsilon}{2H}\right)^{\frac{2}{\alpha}}}\right)$, and it is apparent that the bound we obtain is significantly improved.

## 6 CVaR-sensitive bandits

The concentration bound for CVaR estimation in Proposition 1 opens avenues for bandit applications. We illustrate this claim by using the regret minimization framework in a stochastic $K$-armed bandit problem, with an objective based on CVaR. While CVaR optimization has been considered in a bandit

setting in the literature (cf. Galichet et al. [2013]), the underlying arms' distributions there have bounded support. We relax this assumption, and consider the case of sub-Gaussian distributions for the $K$ arms. The tail bounds in Kolla et al. [2019b] and Kolla et al. [2019a] do not allow a bandit application, because forming the confidence term (required for UCB-type algorithms) using their bound would require knowledge of the density in a neighborhood of the true VaR. In contrast, the constants in our bounds depend only on the sub-Gaussian parameter $\sigma$, and several classic MAB algorithms (including UCB) assume this information.

Suppose we are given $K$ arms with unknown distributions $P_i, i = 1, \ldots, K$. The interaction of the bandit algorithm with the environment proceeds, over $n$ rounds, as follows: (i) select an arm $I_t \in \{1, \ldots, K\}$; (ii) Observe a sample cost from the distribution $P_{I_t}$ corresponding to the arm $I_t$.

Let $C_\alpha(i)$ denote the CVaR, with confidence $\alpha \in (0, 1)$, of the distribution $P_i$ corresponding to arm $i$, for $i = 1, \ldots, K$. Let $C_* = \min_{i=1,\ldots,K} C_\alpha(i)$ denote the lowest CVaR among the $K$ distributions, and $\Delta_i = (C_\alpha(i) - C_*)$ denote the gap in CVaR values of arm $i$ and that of the best arm.

The classic objective in a bandit problem is to find the arm with the lowest expected value. We consider an alternative formulation, where the goal is to find the arm with the lowest CVaR. Using the notion of regret, this objective is formalized as follows:
$$R_n = \sum_{i=1}^K C_\alpha(i) T_i(n) - n C_* = \sum_{i=1}^K T_i(n) \Delta_i,$$
where $T_i(n) = \sum_{t=1}^n \mathbb{I}\{I_t = i\}$ is the number of pulls of arm $i$ up to time instant $n$.

Next, we present a straightforward adaptation CVaR-LCB of the well-known UCB algorithm [Auer et al., 2002] to handle an objective based on CVaR. The algorithm plays each arm once in the initialization phase, and in each of the remaining rounds $t = K + 1, \ldots, n$, plays the arm, say $I_t$, with the lowest UCB value, that is, $I_t = \arg\min_{i=1,\ldots,K} \text{LCB}_t(i)$ with

$$\text{LCB}_t(i) = c_{i,T_i(t-1)} - \frac{2}{1-\alpha} \sqrt{\frac{\log(Ct)}{c\, T_i(t-1)}},$$

where $c_{i,T_i(t-1)}$ is the empirical CVaR for arm $i$ computed using (6) from $T_i(t-1)$ samples, and $C, c$ are constants that depend on the sub-Gaussianity parameter $\sigma$ (see Corollary 1).

The result below bounds the regret of CVaR-LCB algorithm, and the proof is a straightforward adaptation of that used to establish the regret bound of the regular UCB algorithm in [Auer et al., 2002] (see Bhat and Prashanth [2019] for details).

**Theorem 1.** *For a $K$-armed stochastic bandit problem where the the arms' distributions are sub-Gaussian with parameter $\sigma = 1$, the regret $R_n$ of CVaR-LCB satisfies*

$$\mathbb{E}(R_n) \leq \sum_{\{i:\Delta_i>0\}} \frac{16\log(Cn)}{(1-\alpha)^2 \Delta_i} + K\left(1 + \frac{\pi^2}{3}\right)\Delta_i.$$

*Further, $R_n$ satisfies the following bound that does not scale inversely with the gaps:*

$$\mathbb{E}(R_n) \leq \frac{8}{(1-\alpha)}\sqrt{Kn\log(Cn)} + \left(\frac{\pi^2}{3} + 1\right)\sum_i \Delta_i.$$

## 7 Conclusions

We used finite sample bounds from Fournier and Guillin [2015] for the Wasserstein distance between the empirical and true distributions of a random variable to derive two-sided concentration bounds for the error between the true and empirical CVaR, spectral risk measure and CPT-value of a random variable. Our bounds hold for random variables that either have finite exponential moment, or finite higher-order moment, and specialize nicely to sub-Gaussian and sub-exponential random variables. The bound further improves upon previous similar results, which either gave one-sided bounds, or applied only to bounded random variables. In addition, to illustrate the usefulness of our concentration bounds, we used our CVaR concentration bound to provide a regret-bound analysis for an algorithm for a bandit problem where the risk measure to be optimized is CVaR.

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
