[Supplementary Material]



# A Convergence proofs

**Lemma 2.** *Suppose $X$ and $Y$ are r.v.s having CDFs $F_1$ and $F_2$, respectively. Then,*

$$\sup |\mathbb{E}(f(X) - \mathbb{E}(f(Y))| = W_1(F_1, F_2) = \int_{-\infty}^{\infty} |F_1(s) - F_2(s)| \mathrm{d}s = \int_0^1 |F_1^{-1}(\beta) - F_2^{-1}(\beta)| \mathrm{d}\beta, \tag{14}$$

*where the supremum in (2) is over all functions $f : \mathbb{R} \to \mathbb{R}$ that are 1-Lipschitz.*

*Proof.* The first equality in (14) is given by the Kantorovich-Rubinstein theorem (see [Givens and Shortt, 1984, Edwards, 2011]). The second equality is given in [Vallander, 1974].

To prove the third inequality in (14), we note that the integral on the left hand side of the third inequality is unchanged if we replace $F_1$ and $F_2$ by the pointwise maximum and minimum, respectively, of $F_1$ and $F_2$. Hence, without loss of generality, we may assume that $F_1(s) \geq F_2(s)$ for all $s \in \mathbb{R}$. The integral in question then reduces to

$$\int_{-\infty}^{\infty} |F_1(s) - F_2(s)| \mathrm{d}s = \int_{-\infty}^{\infty} (F_1(s) - F_2(s)) \mathrm{d}s = \int_{-\infty}^{\infty} \int_{F_2(s)}^{F_1(s)} \mathrm{d}\beta \mathrm{d}s. \tag{15}$$

It can easily be shown from the definition of the generalized inverse that

$$\begin{aligned} \{(\beta, s) \in \mathbb{R}^2 : F_2(s) < \beta < F_1(s)\} &\subseteq \{(\beta, s) \in \mathbb{R}^2 : F_1^{-1}(\beta) \leq s \leq F_2^{-1}(\beta)\} \\ &\subseteq \{(\beta, s) \in \mathbb{R}^2 : F_2(s) \leq \beta \leq F_1(s)\}. \end{aligned}$$

This justifies interchanging the order of integration (see Theorem 14.14 of Apostol [1974]) in (15), which yields

$$\int_{-\infty}^{\infty} |F_1(s) - F_2(s)| \mathrm{d}s = \int_0^1 \int_{F_1^{-1}(\beta)}^{F_2^{-1}(\beta)} \mathrm{d}s \mathrm{d}\beta = \int_0^1 [F_2^{-1}(\beta) - F_1^{-1}(\beta)] \mathrm{d}\beta. \tag{16}$$

The third inequality in (14) now follows by noting that, under our assumption that $F_1(s) \geq F_2(s)$ for all $s \in \mathbb{R}$, we have $F_2^{-1}(\beta) \geq F_1^{-1}(\beta)$ for all $\beta \in [0, 1]$. $\qquad \square$

## A.1 Proof of Proposition 1

*Proof.* Consider the event $A = \{W_1(F_n, F) \leq (1 - \alpha)\epsilon\}$, where $F_n$ is as defined in (5). Lemma 1 provides a lower bound on $\mathbb{P}(A)$ depending on whether the r.v.s satisfy (C1) or (C2). In particular, we have

$$\mathbb{P}(A) \geq 1 - B(n, (1 - \alpha)\epsilon), \tag{17}$$

where $B(\cdot, \cdot)$ is as defined in Lemma 1.

Applying Lemma 2, we have on the event $A$,

$$\left| \int_{\mathbb{R}} f(x) dF(x) - \int_{\mathbb{R}} f(x)) dF_n(x) \right| \leq (1 - \alpha)\epsilon, \tag{18}$$

for any 1-Lipschitz function $f : \mathbb{R} \to \mathbb{R}$.

Choose $\xi \in \mathbb{R}$ arbitrarily and let $g_\xi(x) = (1 - \alpha)\xi + (x - \xi)^+$. Then,

$$\int_{\mathbb{R}} g_\xi(x) dF(x) = (1 - \alpha)\xi + \mathbb{E}(X - \xi)^+ \triangleq D(\xi), \text{ and}$$

$$\int_{\mathbb{R}} g_\xi(x) dF_n(x) = (1 - \alpha)\xi + \frac{1}{n} \sum_{i=1}^n (X_i - \xi)^+ \triangleq D_n(\xi).$$

Observing that $g_\xi$ is 1-Lipschitz in $x$ for every $\xi \in \mathbb{R}$ and using (18), we obtain

$$|D(\xi) - D_n(\xi)| \leq (1 - \alpha)\epsilon, \text{ on } A, \text{ for any } \xi \in \mathbb{R}.$$

Choose $m > 0$ arbitrarily, and let $\xi_1, \xi_2 \in \mathbb{R}$ be such that

$$D(\xi_1) \leq \inf_\xi D(\xi) + \frac{1}{m}, \text{ and } D_n(\xi_2) \leq \inf_\xi D_n(\xi) + \frac{1}{m}.$$

Then, on the event $A$, we have

$$- (1 - \alpha)\epsilon - \frac{1}{m} \leq D(\xi_1) - D_n(\xi_1) - \frac{1}{m} \leq \inf_\xi D(\xi) - \inf_\xi D_n(\xi)$$

$$\leq D(\xi_2) - D_n(\xi_2) + \frac{1}{m} \leq (1 - \alpha)\epsilon + \frac{1}{m}.$$

Since the chain of inequalities above hold for any $m > 0$, we conclude that

$$\left| \inf_\xi D(\xi) - \inf_\xi D_n(\xi) \right| \leq (1 - \alpha)\epsilon, \text{ on } A. \tag{19}$$

Notice that, by definition, $\inf_\xi D(\xi) = (1 - \alpha)C_\alpha(X)$ and $\inf_\xi D_n(\xi) = (1 - \alpha)C_n$. Thus,

$$|C_\alpha(X) - C_n| \leq \epsilon, \text{ on the event } A.$$

The main claim now follows by using the bound on $\mathbb{P}(A)$ in (17). $\qquad\square$

## A.2 Proof of Proposition 2

*Proof.* Consider the event $A = \{W_1(F_n, F) \leq \epsilon/K\}$, where $F_n$ is as defined in (5). Lemma 1 provides a lower bound on $\mathbb{P}(A)$ depending on whether the r.v.s satisfy (C1) or (C2). In particular, we have

$$\mathbb{P}(A) \geq 1 - B(n, \epsilon/K), \tag{20}$$

where $B(\cdot, \cdot)$ is as defined in Lemma 1.

Equation (10) implies that $A \subseteq \{|m_{n,\phi} - M_\phi(X)| \leq \epsilon\}$. The main claim now follows by using the bound on $\mathbb{P}(A)$ in (20). $\qquad\square$

## A.3 Proof of Proposition 3

*Proof.* Let

$$\Delta_n^+ = \int_0^\infty w^+ \left( \mathbb{P}\left(u^+(X) > z\right)\right) dz - \int_0^\infty w^+ \left(1 - \hat{F}_n^+(z)\right) dz. \tag{21}$$

The quantity above is the difference between the first integral in CPT-value estimate (13) and the first integral in the CPT-value (11). Using (C3), we have

$$\left|\Delta_n^+\right| \leq L \int_0^\infty |F^+(z) - \hat{F}_n^+(z)|^\alpha dz, \tag{22}$$

where $F^+(\cdot)$ is the CDF of the r.v. $u^+(X)$.

Recall that the r.v. $u^+(X)$ is bounded a.s. in $[0, u^+(T_2)]$ by our assumptions on $u^+$ and $X$. Applying Jensen's inequality to the concave $x \mapsto x^\alpha$ after normalizing the Lebesgue measure on the interval $[0, u^+(T_2)]$, we obtain

$$\frac{1}{u^+(T_2)} \int_0^{u^+(T_2)} |F^+(z) - \hat{F}_n^+(z)|^\alpha dz \leq \left[\frac{1}{u^+(T_2)} \int_0^{u^+(T_2)} |F^+(z) - \hat{F}_n^+(z)| dz\right]^\alpha$$

$$\leq \left[\frac{1}{u^+(T_2)} \int_{-\infty}^\infty |F^+(z) - \hat{F}_n^+(z)| dz\right]^\alpha.$$

Applying the second equality in Lemma 2 to the CDFs $F^+$ and $\hat{F}_n^+$ gives

$$\int_0^{u^+(T_2)} |F^+(z) - \hat{F}_n^+(z)|^\alpha dz \leq [W_1(F^+, \hat{F}_n^+)]^\alpha [u^+(T_2)]^{1-\alpha}.$$

379 Using the bound obtained above in (22), we obtain

$$\left|\Delta_n^+\right| \le L[W_1(F^+, \hat{F}_n^+)]^\alpha [u^+(T_2)]^{1-\alpha}.$$

380 Next, for any $\epsilon > 0$, consider the event $A = \{W_1(F^+, \hat{F}_n^+) \le [\epsilon/\{2L[u^+(T_2)]^{1-\alpha}\}]^{1/\alpha}\}$. Then,
381 from Lemma 1,

$$\mathbb{P}(A) \ge 1 - B(n, [\epsilon/\{2L[u^+(T_2)]^{1-\alpha}\}]^{1/\alpha}),$$

382 where $B$ is as given in Lemma 1. On the event $A$, we have $|\Delta_n^+| \le \epsilon/2$.

383 Along similar lines, letting $\Delta_n^- = \int_0^\infty w^- \left(\mathbb{P}\left(u^-(X) > z\right)\right) dz - \int_0^\infty w^- \left(1 - \hat{F}_n^-(z)\right) dz$, it is
384 easy to infer that

$$\left|\Delta_n^-\right| \le \epsilon/2 \text{ on the set } A' = \{W_1(F^-, \hat{F}_n^-) \le [\epsilon/\{2L[u^-(T_1)]^{1-\alpha}\}]^{1/\alpha}\}, \tag{23}$$

385 where $F^-(\cdot)$ is the CDF of $u^-(X)$. The main claim follows by using triangle inequality, that is,

$$
\begin{aligned}
\mathbb{P}(|C_n - C(X)| > \epsilon) &\le \mathbb{P}\left(\left|\Delta_n^+\right| > \epsilon/2\right) + \mathbb{P}\left(\left|\Delta_n^-\right| > \epsilon/2\right) \\
&\le [1 - \mathbb{P}(A)] + [1 - \mathbb{P}(A')] \\
&\le B(n, [\epsilon/\{2L[u^+(T_2)]^{1-\alpha}\}]^{1/\alpha}) + B(n, [\epsilon/\{2L[u^-(T_1)]^{1-\alpha}\}]^{1/\alpha}) \\
&\le 2B(n, [\epsilon/\{2LT^{1-\alpha}\}]^{1/\alpha}).
\end{aligned}
$$

386 This completes the proof. $\qquad\square$

### A.4 Proof of Proposition 4

388 *Proof.* For some positive $\tau_n$ to be specified later, we have

$$
\begin{aligned}
\Delta_n^+ &= \int_0^\infty w^+ \left(\mathbb{P}\left(u^+(X) > z\right)\right) dz - \int_0^{\tau_n} w^+ \left(1 - \hat{F}_n^+(z)\right) dz \\
&= \int_0^\infty w^+ \left(1 - F^+(z)\right) dz - \int_0^{\tau_n} w^+ \left(1 - F^+(z)\right) dz \\
&\quad + \int_0^{\tau_n} w^+ \left(1 - F^+(z)\right) dz - \int_0^{\tau_n} w^+ \left(1 - \hat{F}_n^+(z)\right) dz \\
&= I_1 + I_2,
\end{aligned}
$$

389 where

$$I_1 = \int_{\tau_n}^\infty w^+ \left(1 - F^+(z)\right) dz, \quad I_2 = \int_0^{\tau_n} w^+ \left(1 - F^+(z)\right) dz - \int_0^{\tau_n} w^+ \left(1 - \hat{F}_n^+(z)\right) dz.$$

390 For handling the first term in the RHS above, we start with the following observation:

$$
\begin{aligned}
I_1 = \int_{\tau_n}^\infty w \left(\mathbb{P}\left(u^+(X) > z\right)\right) dz &\le L \int_{\tau_n}^\infty \left(\mathbb{P}\left(u^+(X) > z\right)\right)^\alpha dz \le 8L \int_{\tau_n}^\infty \frac{z}{\tau_n} \exp(-\alpha z^2/2) dz \\
&= \frac{8L}{\tau_n} \frac{2}{\alpha} \exp(-\alpha \tau_n^2),
\end{aligned}
$$

391 where we used the following facts: (i) $w$ is Hölder continuous; (ii) $w(0) = 0$; and (iii) a tail bound
392 for the sub-Gaussian r.v. $u^+(X)$.

393 The second term, i.e., $I_2$ is bounded as follows:

$$\mathbb{P}(I_2 > \epsilon) \le B\left(n, \left(\frac{\epsilon}{L\tau_n^{(1-\alpha)}}\right)^{1/\alpha}\right) = C \exp(-\frac{cn\epsilon^{2/\alpha}}{\tau_n^{2/\alpha}}).$$

$$\text{Or, equivalently } I_2 \le \tau_n \left(\frac{\log(C/\delta)}{cn}\right)^{\alpha/2} \text{ w.p. } 1 - \delta.$$

where the inequality above follows by applying Proposition 3 to the r.v. $Z = \max(u^+(X), \tau_n)$, which takes values in the bounded interval $[0, \tau_n]$. Using the bounds on $I_1$ and $I_2$, w.p. $1 - \delta$, we have

$$\Delta_n^+ \leq \frac{16L}{\alpha\tau_n} \exp(-\alpha\tau_n^2) + \tau_n \left( \frac{\log(C/\delta)}{cn} \right)^{\alpha/2}. \tag{24}$$

Setting $\tau_n = \sqrt{\frac{1}{2} \log n}$, we obtain

$$\Delta_n^+ \leq \frac{16L}{\alpha n^{\alpha/2}} + \sqrt{\frac{\log n}{2}} \left( \frac{\log(C/\delta)}{cn} \right)^{\alpha/2} \quad \text{w.p. } 1 - \delta,$$

leading to

$$\mathbb{P}\left(\Delta_n^+ > \epsilon\right) \leq C \exp\left( -cn \left( \frac{2}{\log n} \right)^{\frac{1}{\alpha}} \left( \epsilon - \frac{16L}{\alpha n^{\alpha/2}} \right)^{\frac{2}{\alpha}} \right).$$

The main claims by inferring a bound similar to the above for the second integrals in $C_n$ and $C(X)$ and then, using a triangle inequality as in the proof of Proposition 3. $\square$

# B  Proof of Theorem 1

*Proof.* The proof follows by using arguments analogous to that in the proof of Theorem 1 in [Auer et al., 2002]. For the sake of completeness, we provide the complete proof.

Let 1 denote the optimal arm, without loss of generality. We bound the number of pulls $T_i(n)$ of any suboptimal arm $i \neq 1$. Fix a round $t \in \{1, \dots, n\}$ and suppose that a sub-optimal arm $i$ is pulled in this round. Then, we have

$$c_{i, T_i(t-1)} - \frac{2}{(1-\alpha)} \sqrt{\frac{\log(Ct)}{c\, T_i(t-1)}} \leq c_{1, T_1(t-1)} - \frac{2}{(1-\alpha)} \sqrt{\frac{\log(Ct)}{c\, T_1(t-1)}}. \tag{25}$$

The LCB-value of arm $i$ can be larger than that of 1 *only if* one of the following three conditions holds:

**(1)** $c_{1, T_1(t-1)}$ **is outside the confidence interval:**

$$c_{1, T_1(t-1)} - \frac{2}{(1-\alpha)} \sqrt{\frac{\log(Ct)}{c\, T_1(t-1)}} > C_\alpha(1), \tag{26}$$

**(2)** $c_{i, T_i(t-1)}$ **is outside the confidence interval:**

$$c_{i, T_i(t-1)} + \frac{2}{(1-\alpha)} \sqrt{\frac{\log(Ct)}{c\, T_i(t-1)}} < C_\alpha(i), \tag{27}$$

**(3) Gap $\Delta_i$ is small:** If we negate the two conditions above and use (25), then we obtain

$$C_\alpha(i) - \frac{4}{(1-\alpha)} \sqrt{\frac{\log(Ct)}{c\, T_i(t-1)}} \leq c_{i, T_i(t-1)} - \frac{2}{(1-\alpha)} \sqrt{\frac{\log(Ct)}{c\, T_i(t-1)}}$$

$$\leq c_{1, T_1(t-1)} - \frac{2}{(1-\alpha)} \sqrt{\frac{\log(Ct)}{c\, T_1(t-1)}} \leq C_\alpha(1)$$

$$\Rightarrow \quad \Delta_i < \frac{4}{(1-\alpha)} \sqrt{\frac{\log(Ct)}{c\, T_i(t-1)}} \text{ or } T_i(t-1) \leq \frac{16 \log(Ct)}{(1-\alpha)^2 \Delta_i^2} \tag{28}$$

Let $u = \dfrac{16 \log(Cn)}{(1-\alpha)^2 \Delta_i^2} + 1$. When $T_i(t-1) \geq u$, i.e., when the condition in (28) does not hold, then either (i) arm $i$ is not pulled at time $t$, or (ii) (26) or (27) occurs. Thus, we have

$$
T_i(n) = 1 + \sum_{t=K+1}^{n} \mathbb{I}\{I_t = i\}
$$

$$
\leq u + \sum_{t=u+1}^{n} \mathbb{I}\{I_t = i; T_i(t-1) \geq u\}
$$

$$
\leq u + \sum_{t=u+1}^{n} \mathbb{I}\left\{ c_{i,T_i(t-1)} - \frac{2}{(1-\alpha)}\sqrt{\frac{\log(Ct)}{c\, T_i(t-1)}} \right.
$$

$$
\left. \leq c_{1,T_1(t-1)} - \frac{2}{(1-\alpha)}\sqrt{\frac{\log(Ct)}{c\, T_1(t-1)}}; T_i(t-1) \geq u \right\}
$$

$$
\leq u + \sum_{t=1}^{\infty}\sum_{s=1}^{t-1}\sum_{s_i=u}^{t-1} \mathbb{I}\left\{ C_i(s_i) - \frac{2}{(1-\alpha)}\sqrt{\frac{\log(Ct)}{c\, s_i}} \leq C_1(s) - \frac{2}{(1-\alpha)}\sqrt{\frac{\log(Ct)}{c\, s}} \right\}
$$

$$
\leq u + \sum_{t=1}^{\infty}\sum_{s=1}^{t-1}\sum_{s_i=u}^{t-1} \mathbb{I}\left\{ \left( C_\alpha(1) < C_1(s) - \frac{2}{(1-\alpha)}\sqrt{\frac{\log(Ct)}{c\, s}} \right) \right.
$$

$$
\left. \text{or } \left( C_\alpha(i) > C_i(s_i) + \frac{2}{(1-\alpha)}\sqrt{\frac{\log(Ct)}{c\, s_i}} \right) \text{ occurs} \right\}.
$$

Using Proposition 1, we can bound the probability of occurrence of each of the two events inside the indicator on the RHS of the final display above as follows:

$$
\mathbb{P}\left( C_\alpha(1) < C_1(s) - \frac{2}{(1-\alpha)}\sqrt{\frac{\log(Ct)}{c\, s}} \right) \leq \frac{1}{t^4}, \text{ and}
$$

$$
\mathbb{P}\left( C_\alpha(i) > C_i(s_i) + \frac{2}{(1-\alpha)}\sqrt{\frac{\log(Ct)}{c\, s_i}} \right) \leq \frac{1}{t^4}.
$$

Plugging the bounds on the events above and taking expectations on $T_i(n)$ related inequality above, we obtain

$$
\mathbb{E}[T_i(n)] \leq u + \sum_{t=1}^{\infty}\sum_{s=1}^{t-1}\sum_{s_i=u}^{t-1} \frac{2}{t^4} \leq u + 2\sum_{t=1}^{\infty}\frac{1}{t^2} \leq u + \frac{\pi^2}{3}. \tag{29}
$$

The preceding analysis together with the fact that $\mathbb{E}R_n = \sum_{i=1}^{K} \Delta_i \mathbb{E}[T_i(n)]$ leads to the first regret bound presented in the theorem.

For inferring the second bound on the regret, i.e., the bound that does not scale inversely with the gaps, observe that

$$
\mathbb{E}R_n = \sum_i \Delta_i\, \mathbb{E}[T_i(n)] = \sum_{i:\Delta_i \leq \lambda} \Delta_i\, \mathbb{E}[T_i(n)] + \sum_{i:\Delta_i \geq \lambda} \Delta_i\, \mathbb{E}[T_i(n)], \text{ for } \lambda > 0
$$

$$
\leq n\lambda + \sum_{i:\Delta_i \geq \lambda}\left( \frac{16 \log(Cn)}{(1-\alpha)^2 \Delta_i} + \Delta_i\left(\frac{\pi^2}{3} + 1\right) \right), \text{ (Using (29) and } \sum_{i:\Delta_i \leq \lambda} \mathbb{E}[T_i(n)] \leq n)
$$

$$
\leq n\lambda + \left( \frac{16K \log(Cn)}{(1-\alpha)^2 \lambda} \right) + \left(\frac{\pi^2}{3} + 1\right)\sum_i \Delta_i,
$$

$$
\leq \frac{8}{(1-\alpha)}\sqrt{Kn \log(Cn)} + \left(\frac{\pi^2}{3} + 1\right)\sum_i \Delta_i, \quad \left( \text{Using } \lambda = \frac{8\sqrt{K \log(Cn)}}{(1-\alpha)} \right).
$$

$\square$