[Reviews · NeurIPS 2019]

Reviewer 1



Originality: the main sources of originality here appear to be Prop 2, as well as the use of properties of the Wasserstein distance to get sharper results (across all the contributions); I have some more comments on this point, later on. Quality: the quality seems good. I skimmed the proofs and they made sense. Clarity: the paper is reasonably well-written; I have some more comments on this point, later on. Significance: the paper seems like a good step forward in terms of giving concentration inequalities for spectral risk measures. The main result(s) are not all that surprising, and the math involved behind-the-scenes doesn't appear to be exceptionally difficult, but the results still seem valuable/worthwhile.

Reviewer 2



This paper is clearly written and easy to read. However, I have some questions about the contribution of this paper. I’m wondering why it is a significant problem to evaluate the estimation error by the Wasserstein distance. While the CVaR is a very important measure, it is no clear connection with the characteristics of the Wasserstein distance. The authors just mention “it is interesting to know” (line 38), but it is very insufficient to describe the importance of their goal. Also, technical contributions seem less. With the known formulation of the CVaR in (6), the bound for the error is obtained by a simple application of the concentration inequality of sub-Gaussian random variables. To improve technical contributions, the authors should investigate theoretical aspects of the error bound such as a convergence rate, adaptivity and so on. ====== UPDATED AFTER REBUTTAL ====== Thanks to the rebuttal by the authors, I understood that there are some connections between the risk measure and the Wasserstein distance. Thus, I updated my score from 5 to 6. However, I feel that it is not easy task for readers to understand the connection from the main script. Though it is always difficult to describe all in short space of conference papers, I'll be glad if more intuitive description is provided.

Reviewer 3



The key insight of the proposed method is to put the estimation of the error in relation to the distance between the empirical and the true distribution, where the distance is the Wasserstein distance. This is a pretty original approach, as far as this reviewer knows. The page limit of NeurIPS puts highly-theoretical papers such as this one at a serious disadvantage. As a result, it is quite hard for someone not expert in the area to grasp what is going on in the paper because very little guidance is given (as there is no space for it), and very little intuition is successfully communicated to the reader. While it is very hard to achieve given the space limitation, perhaps some nuggets of intuition can be made evident here and there. The contribution of the paper seem quite significant, and the technique very novel, although there seems to be a very strong reliance on existing concentration bounds. It would be helpful to clarify how much of the proposed results are more than an application of existing bounds (which would still make the work an interesting contribution, but this clarification would help better assessing its significance).

[Author Response · NeurIPS 2019]

1 We would like to thank the reviewers for their comments.

## Reviewer 1

**On suggested improvements to Sec 6:** We agree that an empirical evaluation would be interesting. However, given that the thrust of this work has been to lay the foundations necessary for employing risk measures in a ML-type application (for instance, the CVaR-bandits application), and coupled with space limitations, we chose to postpone a detailed empirical investigation to future work. Nevertheless, we shall add numerical experiments in a longer version of this paper, and make it available on arxiv.

**On CPT result being out of place:** We believe that CPT is a general risk measure, and our result on CPT-concentration ties in well with the overall theme of the paper, which is to obtain concentration bounds for risk measures, and use Wasserstein distance along the way to achieve the result. The bounds we obtain for the case of CPT improve on the state-of-the-art, esp. for the sub-Gaussian case. As an aside, one can choose an appropriate weight function in the definition of CPT-value, and recover CVaR.

**On the source of sharper results:** We obtain sharper bounds by 1) relating the estimation error directly to one of the three characterizations given in eq. (2) of the Wasserstein distance between the empirical and true distributions, and 2) using concentration bounds for the Wasserstein distance between the true and empirical distribution given in Fournier and Guillin (2015).

**On why Wasserstein metric was used:** The estimation errors for all three risk measures can be directly related to the Wasserstein distance (this is clear from the proofs, but we will also clarify this in the main body of the final paper). Moreover, the bounds resulting from its use apply to unbounded random variables as well. If the DWK inequality were to be used to obtain concentration bounds for CVaR/CPT, the sup norm in the DKW inequality will make the resulting bounds applicable only to bounded r.v.s.

**On how sharp are our results:** We believe our bounds for the sub-Gaussian case are the best achievable in a minimax sense. Setting alpha=0, we recover the expected value, and it is apparent that the dependence on number of samples n and accuracy epsilon cannot be any better. On the other hand, for the case of sub-exponential or even the case of distributions with bounded higher moments, our bounds could be improved. As remarked on l. 153-159, the source of sub-optimality is not in our analysis. Instead, the Wasserstein concentration result from [Fournier and Guillin, 2015] for the latter case is far from optimal, as it involves a power law decay instead of an exponential one.

## Reviewer 2

**On the connection between CVaR and the Wasserstein distance:** As observed by reviewer 3, the estimation error in each of the three risk measures that we consider is directly related to one of the three characterizations of Wasserstein distance given in eq. (2). For instance, as shown in the proof of Prop. 1, the estimation error in CVaR is related to the first equality in (2). Similarly, the estimation errors in CPT and spectral risk measures are related to the 2nd and 3rd equalities in (2).

**On the bound being simple application of a sub-Gaussian concentration result** We strongly disagree. In fact, sub-Gaussian concentration bounds (for e.g., Hoeffding's inequality) is never invoked in our proofs. Instead, we relate the CVaR estimation error to the distance between empirical and true distributions, and then invoke an inequality from [Fournier and Guillin, 2015]. The concentration bound that we derive is a convergence rate result, as it shows that the CVaR estimate converges at an exponential rate to the true CVaR.

**On the strength of the technical contributions** As outlined in l. 43-60, our bounds show a much better dependence on the number of samples $n$ and accuracy $epsilon$, as compared to the state-of-the-art. Moreover, unlike our approach, an alternative proof that is based on quantiles (cf. [Kolla et al. 2019]) does not allow a bandit application. Finally, our approach is unified in the sense that one does not require separate proofs to handle the cases of sub-Gaussian, sub-exponential, or even distributions with bounded higher moments.

## Reviewer 3

**On the reliance on existing results** While the estimation error for each risk measure is related to the Wasserstein distance between EDF and CDF, the proof takes a different route in arriving at this relation for CVaR, spectral risk measures, and CPT. In the first case, we use Lipschitz-ness and inf-norm for CVaR, while spectral risk measures are handled by a relation involving distribution inverses. Finally, for the case of CPT, when the distribution has bounded support, we can relate the estimation error to Wasserstein distance; the case of sub-Gaussian distribution involves more work, as we employ a truncated CPT-value estimator, and show that for such a scheme, one can handle the truncated and non-truncated part separately in the proof.

[Meta-Review · NeurIPS 2019]

The paper is concerned with deriving concentration bounds for measures of 'risk' (e.g. the risk of a portfolio) in financial applications, e.g. CVar, and generalizations thereof, some of which can roughly be viewed as expected deviations from quantiles of a distribution. The main contribution is to remark that changes in risks over two distributions can be bounded through the (l1) Wasserstein distance. Concentration bounds for Wasserstein distance can then be applied. The approach yields 2-sided concentration bounds (for empirical version of such measures) which apparently were not previously obtained, or obtained under stronger boundedness conditions. Authors also discuss achieving 'tighter' bounds than previous one-sided bounds under tail conditions (e..g Gaussian tails), but however do not provide those previous bounds for comparison in neither the main paper nor the rebuttal. In summary, the paper is interesting in deriving a different approach for obtaining concentration results for complex functionals of a distribution, but however might not necessarily be of interest to a large ML audience given the narrow focus in Finance.